# Sera from Patients with NMOSD Reduce the Differentiation Capacity of Precursor Cells in the Central Nervous System

**DOI:** 10.3390/ijms22105192

**Published:** 2021-05-14

**Authors:** Ulises Gómez-Pinedo, Yolanda García-Ávila, Lucía Gallego-Villarejo, Jordi A. Matías-Guiu, María Soledad Benito-Martín, Noelia Esteban-García, Inmaculada Sanclemente-Alamán, Vanesa Pytel, Lidia Moreno-Jiménez, Francisco Sancho-Bielsa, Lucía Vidorreta-Ballesteros, Paloma Montero-Escribano, Jorge Matías-Guiu

**Affiliations:** 1Laboratory of Neurobiology, Department of Neurology, Institute of Neurosciences, San Carlos Health Research Institute, Universidad Complutense, 28040 Madrid, Spain; yoliga08@gmail.com (Y.G.-Á.); lucgallev@gmail.com (L.G.-V.); jordimatiasguiu@hotmail.com (J.A.M.-G.); msbm65@gmail.com (M.S.B.-M.); noeste01@ucm.es (N.E.-G.); inmasancle4@gmail.com (I.S.-A.); vanesa.pytel@gmail.com (V.P.); lidiamor-92@hotmail.com (L.M.-J.); lvidorreta@alumni.unav.es (L.V.-B.); pmontero84@gmail.com (P.M.-E.); matiasguiu@gmail.com (J.M.-G.); 2Department of Physiology, Ciudad Real School of Medicine, Universidad de Castilla-La Mancha, 13001 Ciudad Real, Spain; francisco.sancho@uclm.es

**Keywords:** AQP4–IgG, neurogenesis, neural stem cells, neuromyelitis optica, NMOSD, precursor cells, remyelination

## Abstract

Introduction: AQP4 (aquaporin-4)–immunoglobulin G (IgG)-mediated neuromyelitis optica spectrum disorder (NMOSD) is an inflammatory demyelinating disease that affects the central nervous system, particularly the spinal cord and optic nerve; remyelination capacity in neuromyelitis optica is yet to be determined, as is the role of AQP4–IgG in cell differentiation. Material and Methods: We included three groups—a group of patients with AQP4–IgG-positive neuromyelitis optica, a healthy group, and a sham group. We analyzed differentiation capacity in cultures of neurospheres from the subventricular zone of mice by adding serum at two different times: early and advanced stages of differentiation. We also analyzed differentiation into different cell lines. Results and Conclusions: The effect of sera from patients with NMOSD on precursor cells differs according to the degree of differentiation, and probably affects oligodendrocyte progenitor cells from NG2 cells to a lesser extent than cells from the subventricular zone; however, the resulting oligodendrocytes may be compromised in terms of maturation and possibly limited in their ability to generate myelin. Furthermore, these cells decrease in number with age. It is very unlikely that the use of drugs favoring the migration and differentiation of oligodendrocyte progenitor cells in multiple sclerosis would be effective in the context of neuromyelitis optica, but cell therapy with oligodendrocyte progenitor cells seems to be a potential alternative.

## 1. Introduction

Neuromyelitis optica spectrum disorder (NMOSD) is a rare neuroinflammatory demyelinating disease of the central nervous system (CNS) that mainly attacks the optic nerves and the spinal cord, which can lead to vision loss and paralysis [1,2]. A considerable percentage of patients with NMOSD are positive for IgG1 autoantibodies (IgG) targeting the aquaporin-4 (AQP4) water channel protein expressed in astrocytes (AST); other antibodies may also be detected, such as myelin oligodendrocyte glycoprotein (MOG) antibodies [3,4,5,6]. Other studies have observed anti-vascular endothelial antibodies closely related with the blood–brain barrier (BBB) permeability [7], resulting in an inflammatory response with complement participation, leading to demyelination [8] and, therefore, to neurological sequelae [9]. AQP4–antibody disease has an age of onset of 40 years and shows a high female-to-male ratio (up to 9:1). Today, there is no specific treatment for NMOSD [8,9,10,11]. Treatment is based on the use of immunosuppressants, plasmapheresis, and anti-CD19 and anti-CD20 drugs, aiming to reduce the presence of B cells [12,13]. Recently, new strategies with promising results in preclinical stages are being evaluated, based on AQP inhibitors and anti-AQP4 monoclonal antibodies [11,12]. An important consideration is that remyelination is thought to be limited in NMOSD, and no treatments are able to restore myelin; however, the use of drugs that may favor remyelination is being considered for the treatment of multiple sclerosis [13], and it has also been suggested that they may promote remyelination in NMOSD, as early loss of oligodendrocytes (OLG) has been reported [14,15,16,17]. Other authors have reported that NMO–IgG may decrease the expression of connexins and cause oligodendrocyte damage and demyelination in NMOSD [18]. For instance, clobetasol, which promotes OLG differentiation [19], has been analyzed in an in vivo NMOSD model, with promising results. However, this hypothesis raises unresolved questions about NMOSD: the effect of the altered interaction between ASTs and OLG–ASTs [20,21], whether BBB disruption in NMOSD may inhibit OLG migration, whether the inflammatory environment in active NMOSD lesions and the altered microglial response may inhibit remyelination and favor irreversible axonal lesions [22], and whether AQP4–IgG (present in patient serum) may have an effect on cell differentiation in the central nervous system. In this article, we present our analysis of the latter point. 

## 2. Results

### 2.1. Patients and Controls

*AQP4–IgG detection:* Increased detection of AQP4–IgG was observed in tissue treated with samples from the NMOSD group (high levels of fluorescent signal), although intensity varied between cases; very slight levels were observed in the healthy group, and no labeling was observed in the sham group (Figure 1A) (tissue incubated without any serum). Labeling was observed in all the sections analyzed, regardless of the anatomical region (cortex, cerebellum, hypothalamus, etc.). By location, labeling mainly corresponded to glial cells (astrocytes and astrocyte endfeet) (Figure 1B,C or pericytes surrounding capillaries; Figure 1C, and Appendix A). To confirm whether the presence of NMO–IgG is similar to the presence of anti-AQP4 antibodies in the serum of the patients, a double IHC study was performed. Serum from patients with NMOSD was used as the primary antibody; in parallel and in another tissue, a commercial AQP4 antibody was used instead of patient serum. The GFAP antibody was used in both conditions, with both cases showing very similar labeling and colocalization with astrocytes from the periphery of capillaries (AQP4-positive astrocyte endfeet) and with the walls of the capillaries (Figure 2).

### 2.2. Effect on SVZ Cell Differentiation

This effect was analyzed in two scenarios: in the first, serum was added in late stages of the differentiation culture (protocol 1), and in the second, serum was added at the beginning of the differentiation culture. In the first scenario, we observed an increased proportion of differentiation into ASTs (from 50% to 80%) and a gradual decrease at days 3, 7, and 10 in the percentage of neurons (from 20% to 10%) and OLGs (from 22% to 9%) (*p* < 0.05). All three types of cells presented normal morphology (Figure 3). When serum was added at the beginning of the differentiation culture (protocol 2), gliogenesis was accelerated; an increase was observed in AST labeling (92%), with presence of elongated, fibrous ASTs with long filopodia, and a gradual disappearance of neurons (from 10% to 6%) and OLGs (2%) was also observed (*p* < 0.05). After contact with patient sera, neurospheres showed an aberrant shape, losing their characteristic morphology (Figure 4). In both protocols, an increase in gliogenesis is striking. Another phenomenon observed in both protocols was a decrease in the total number of cells on day 10 of differentiation; in the neurospheres treated with NMOSD patient sera, a 25% decrease in their clonic capacity was observed. The ratio for all NMO cells vs all control cells was 0.76 for protocol 1 and 0.74 for protocol 2.

## 3. Discussion

In recent years, a new function of AQP4 has been described, specifically its role in the inflammatory response, mediating the migration of immune cells, edema, and as a marker of poor prognosis in gliomas [23]. Moreover, a subgroup of AQP has been described that also facilitates transmembrane diffusion of small polar solutes, not just water; these are aquaglyceroporins, with a narrow function in lipid metabolism [23,24,25,26], presenting a slight association with such diseases as diabetes mellitus type 2. 

AQP4 is a key molecule for maintaining water homeostasis in the central nervous system; it is expressed in adult neural stem cells and ASTs, and may be in contact with blood vessels [27]. As it is a transmembrane protein, it is expressed on the surface of cells [28,29]. The presence of a change in the expression of astrocytes may be considered as a therapeutic target. Salman et al. [30], in an in vitro model using human cortical astrocytes, reported that hypothermia can induce an increment of the expression of AQP4; this can be a favorable strategy for the treatment of edema. Recent experimental studies indicate that the localized inhibition of AQP4 can favor hastened functional recovery in lesions such as stroke or traumatic lesions where inflammatory substrate plays an important role [31,32,33]. In mice, AQP4 is expressed in the SVZ, where neural stem cells reside [34], as well as in neural stem cell lines [35]. Furthermore, AQP4 is reported to promote neurogenesis both in the SVZ and in the hippocampus [36,37], and it has, therefore, been suggested to play a role in neurorepair [38] as well as in neurodegenerative diseases, since AQP4 participates in the onset and progression of Alzheimer disease and Parkinson’s disease [39]. AQP4-deficient mice show inhibited neuronal proliferation, migration, and differentiation in adult neuronal cells cultured in vitro [40,41]; we may hypothesize that the presence of AQP4-IgG autoantibodies may result in a similar situation. This situation was observed in both protocols where after ten days, the proliferative capacity decreased by 25%, a fact already observed in knockout models inducing G2-M arrest in neural progenitors in the SVZ [41]. In this sense, the exposure of AQP4 in ASTs to AQP4-IgG autoantibodies causes AQP4 endocytosis, hence the presence of the protein in the cytoplasm. However, this effect may be different in an inflammatory context [42,43], as cytokines also influence neurogenesis [44]. 

Our study shows that differentiation into ASTs decreases when serum is added in the late stages of differentiation cultures, but that differentiation into OLGs and neurons with normal characteristics persists, although to a lesser degree. This implies that AQP4–IgG autoantibodies do not impede the development of OLGs from partially differentiated cells, such as oligodendrocyte progenitor cells (OPCs). However, when autoantibodies are added at the early stages of differentiation, neurogenesis is altered in a similar way to that observed in AQP4-deficient mice, blocking differentiation into OLG. Thus, we may hypothesize that AQP4–IgG has different effects on the differentiation of stem cells and on more immediate precursors (Figure 5).

NMOSD lesions may present limited remyelination due to several mechanisms: inhibition of differentiation into mature OLGs, inhibition of the migration of OPCs through BBB injury, microglial alterations that limit phagocytosis of myelin debris, or axonal alterations. Using recombinant monoclonal IgG1, the loss of OPCs is practically complete in NMOSD, unlike in multiple sclerosis, in which it decreases to 50% [45]; therefore, with the presence of AQP4–IgG, remyelination may be difficult [46]. Our study presents the limitation that it does not reproduce the pathogenesis of NMOSD, which involves an inflammatory environment with BBB disruption, nor the effects of axonal alteration: it only analyzes the effect of AQP4IgG on cell differentiation and does not consider the effect of AQP4 on the immune response [47]. Another possibility that may be directly related to the expression of AQP4–IgG is the involvement of connexins, closely related to cell adhesion molecules, in the possible origin of the irregular shape of the neurospheres. The alterations observed in their shape, as well as the presence of anti-vascular endothelial IgG, or alterations in glutamate cytotoxicity, may influence the loss of oligodendrocytes and their myelinic potential [7,17,18]. 

Remyelination in NMOSD lesions is not well understood [48]. It should be noted that AQP4 is not expressed in OLGs. The use of recombinant monoclonal IgG1 from the cerebrospinal fluid of patients with NMOSD has provided information in in vitro and in vivo models [49], and it has been shown that the OLGs repopulating the lesions are not mature and are unable to remyelinate, unlike in multiple sclerosis. Despite this, we can draw several conclusions. Firstly, NMOSD patient serum decreases neurogenesis, similarly to the effect observed in AQP4-deficient mice. Secondly, NMOSD patient serum affects precursor cells differently, depending on the degree of differentiation, and probably has a less marked effect on OPCs from NG2 cells than on SVZ cells; however, it is possible that the resulting OLGs may be unable to generate myelin. Furthermore, these cells decrease in number with age. AQP4–IgG-positive NMOSD is an autoimmune astrocytopathy, and demyelination occurs only as a consequence of the astrocytic alteration, as in other pathogenic situations [50]. Thirdly, drugs favoring migration and potential subsequent differentiation of OPCs in multiple sclerosis represent a new line of treatment to promote remyelination; however, this type of treatment, although effective, would not be so in NMOSD in this context. To achieve remyelination, cell therapy with OPCs [51], which may protect against AQP4–IgG, seems to be the most probable target in the search for a restorative therapy for NMOSD. At this point, a potential synergy between cell therapy and bioengineering may involve the use of biomaterials with biomedical applications in the central nervous system, such as chitosan [52], which may condition OPCs to avoid the effects of AQP4–IgG and, thus, reach target sites to promote remyelination; this may represent a promising alternative.

Three-dimensional culture techniques based on the generation of organoids derived from induced pluripotent stem cells, developed in recent years, may be of great value in advancing the generation of biohybrid devices to evaluate new therapeutic alternatives, furthering the understanding of molecular transport mechanisms, and providing an individualized model to simulate conditions close to those that could be found in actual patient tissue [53,54]. High-throughput screening (HTS) techniques can be a complementary tool, as well as assays based on cells and biochemical techniques such as those described by Aldewachi et al. [55] in 2021 and by Pérez del Palacio et al. [56] in 2016, which are necessary and fundamental for analyzing the potential of possible drugs.

Our study presents some limitations. Although the data presented show considerable differences between the NMOSD group and the control group, with serum from patients with NMOSD having a clear effect on neural differentiation, we must acknowledge that the sample size is small, and that our serum samples may contain other antibodies or undetermined or unknown molecules; future studies should use larger samples and quantify the antibody in order to establish a correlation with clinical and imaging profiles.

In conclusion, our study shows that NMOSD patient serum has a similar effect in decreasing neurogenesis to that observed in AQP4-deficient mice. NMOSD patient serum affects precursor cells differently depending on the degree of differentiation, and probably has less of an effect on oligodendrocyte progenitor cells from NG2 cells than on SVZ cells; however, the resulting OLGs may be unable to generate myelin. Furthermore, these cells decrease in number with age. It is very unlikely that the use of drugs favoring the migration and differentiation of OPCs in multiple sclerosis would be effective in NMOSD. In future, the use of cell therapy with OPCs seems to be a potential alternative, with progress already being made in the area of demyelinating diseases. Research into the action of AQP4–IgG on central nervous system precursor cells may contribute to the understanding of the demyelination mechanism caused by a primarily astrocytic alteration.

## 4. Material and Methods

### 4.1. Study Groups

We included three study groups: a group of patients with NMOSD (NMOSD group, mean age: 47.62 ± 10.25 years; sex: one male and five female), a group of healthy patients without neurological or immune diseases (healthy group, mean age: 40.67 ± 16.65 years; sex: one male and two female), and a sham group, for which the sample included no serum (only culture medium). All six patients from the NMOSD group met diagnostic criteria for NMOSD [57] and tested positive for AQP4–IgG in measurements taken during clinical diagnosis. The three controls were healthy individuals with no relevant neurological or immunological history. The sham group included samples containing the solvent solution only and no human serum. The study was approved by the Clinical Research Ethics Committee of Hospital Clínico San Carlos, under authorization code 16010E. All participants (cases and controls) agreed to participate in the study and signed informed consent forms. We collected the following information from the NMOSD group: demographic variables (age, sex) and clinical, diagnostic, and therapeutic data related to the disease (personal history, family history, disease onset, diagnostic criteria, progression time, number of episodes, previous and current treatment, optical coherence tomography findings, brain and spinal magnetic resonance imaging findings). The clinical profiles of patients with NMOSD are summarized in Appendix A. 

### 4.2. Experimental Animals

All experiments were conducted at the vivarium of Hospital Clínico San Carlos, where animals were kept under adequate vivarium conditions, with access to water and food ad libitum and behavioral enrichment elements; temperature was maintained at 21 °C (±1 °C), with 12:12 light–dark cycles. The study complied with Directive 2010/63/EU of the European Parliament and of the Council of 22 September 2010 on the protection of animals used for scientific purposes (Official Journal of the European Union, 20 October 2010, L 276/33), which entered into force in Spain in February 2013 with the publication of Royal Decree 53/2013 of 1 February 2013, and was supervised by the local Bioethics and Animal Welfare Committee of our hospital and the Region of Madrid. We used female BALB/c mice of six weeks of age for the extraction of neurospheres and tissue samples in order to analyze AQP4–IgG antibody titers in cases and controls.

### 4.3. Sample Extraction and Storage, and Confirmation of the Presence of AQP4–IgG in Serum Samples and Mouse Cerebella Treated with the Serum

In both groups, samples were obtained during routine diagnostic blood collection. Samples were coded and stored at −80 °C. Presence of AQP4–IgG was determined using an enzyme-linked immunosorbent assay (ELISA; EIASON Aquaporin-4 Ab V2; IASON GmbH, Austria). Using immunohistochemistry (IHC), we confirmed the presence of the antibody by treating mouse brain samples with serum from the NMOSD group and from the healthy group at a 1:25 dilution, followed by a fluorochrome-conjugated antibody (Sigma; Ref. F4512-1ML). To confirm whether the affinity observed in the serum pool was similar to that of a commercial antibody (Millipore AB3594, 1:100) and whether the labeling was specific to astrocytes (Ck GFAP BioLegend 829401, 1:800), we carried out an IHC study and subsequently analyzed samples with a confocal microscope and the Olympus AF1000 microscope software, using a sequential acquisition protocol with a resolution of 1024 px, at 2 ms/px, with a magnification of 40× (NA 0.84); regions of interest for more detailed analysis were identified according to the procedure described by Kim et al. [58] (Appendix A). 

### 4.4. Primary Culture of Neurospheres from the Subventricular Zone

To obtain cells from the subventricular zone (SVZ), we dissected the brains of three adult BALB/c mice of six weeks of age. The primary neurosphere cultures were obtained according to the procedure designed by Belenguer et al. [24] (Appendix A). Neurospheres that were already expanded and free of detritus were used until they became confluent and prevented from adhering or beginning spontaneous differentiation; they reached the optimal point at the third or fourth passage of proliferation.

### 4.5. Effects of the Serum Sample on Differentiation

Disaggregated neurospheres were seeded into Petri dishes coated with 0.01% poly-L-lysine solution to facilitate adhesion and expansion, changing from an initial spherical shape to an elongated form. Differentiation medium 1 was added to the coated dishes; this medium is made up of a control medium supplemented with 1.5% of fetal bovine serum (Gifco; Ref: 16000-036) and a 2% dilution of human serum (combined mix of equal parts from all NMOSD patients or control serum samples), at a density of 25,000 cells in chamber slides. We analyzed two different scenarios (each with their corresponding control). In the first scenario, patients’ serum was added to the neurosphere culture after adding the differentiation medium, which is necessary to induce a period of conditioning of the neurospheres. We also conducted a control experiment, in which we added serum from patients with no associated neurological disease, after adding the differentiation medium. In the second scenario studied, we added serum from patients with NMOSD simultaneously with the differentiation medium, i.e., without respecting the conditioning period for differentiation. As in the first scenario, a control experiment was performed in which serum from patients without associated neurological disease was added at the same time as the differentiation medium. We then analyzed the cultures on days 3, 7, and 10 to determine the time of biological vulnerability after administration of the serum in each scenario (culture media are listed in Appendix A). We assessed the degree of differentiation by performing an immunohistochemical test to analyze the expression of cell markers (GFAP, Olig2, and TUJ1). On each assessment day, chamber slides were fixed with a 4% paraformaldehyde solution and 7% sucrose for 20 min at room temperature. Chamber slides were then washed three times for five minutes with a solution of phosphate buffered saline (PBS), triton, and albumin, and incubated for three hours at room temperature with the mixture of primary antibodies to identify the different cell lines: TUJ1 (Mouse, Abcam 78078, 1:200) for neuron labeling, Olig2 (Rabbit, Abcam 81093, 1:250) for OLG labeling, and GFAP (Chicken, Ab 4674, 1:1000) for AST labeling. Cells were then washed with PBS and incubated for an hour with the corresponding secondary antibodies (Alexa Fluor, Invitrogen), also diluted in PBS. We used specific secondary antibodies for each primary antibody, conjugated to a fluorophore for confocal microscopy analysis, and 0.2 mg/mL of 4′,6-diamidino-2-phenylindole (1:2000 dilution) to visualize nuclei. Slides were then washed with PBS, and Fluorsave^®^ mounting medium was added for subsequent analysis at several magnifications with the Olympus AF 1000 confocal microscope.

### 4.6. Statistical Analysis

Samples were processed in triplicate, and two independent experiments were performed for each scenario. Cells in up to five chamber slides were counted using the Olympus Ix83 fluorescence microscope at 40× or 63× magnification; more than 250 cells were counted for each chamber slide. Statistical analysis was performed using StatView 5.0 and GraphPad Prism software. The absolute values obtained were analyzed using one-way analysis of variance followed by a Tukey test for multiple comparisons. Graphs show the averages obtained from the analyzed values for each group. All values are expressed as means ± SE. Statistical significance was set at *p* < 0.05.

## Figures and Tables

**Figure 1 ijms-22-05192-f001:**
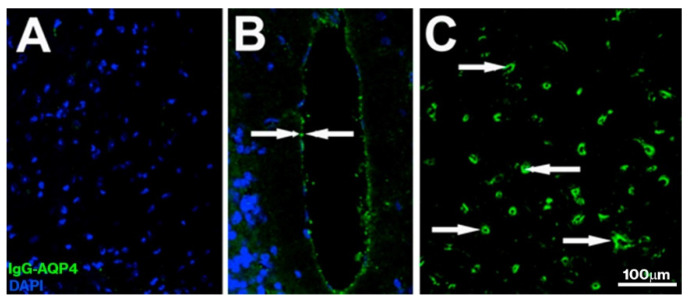
Confirmation of the presence of AQP4–IgG in mouse brain tissue. Representative images of AQP4–IgG are shown. Panel (**A**) shows a characteristic image from the healthy group; no labeling was observed in the sham group (image not shown). Panels (**B**,**C**) show representative images of samples treated with serum from patients with AQP4–IgG-positive NMOSD. In the NMOSD group, labeling was observed in cells adjacent to the third ventricle (B, arrows) and pericytes surrounding capillaries of the cerebral cortex (**C**, arrows). Scale bar = 100 μm.

**Figure 2 ijms-22-05192-f002:**
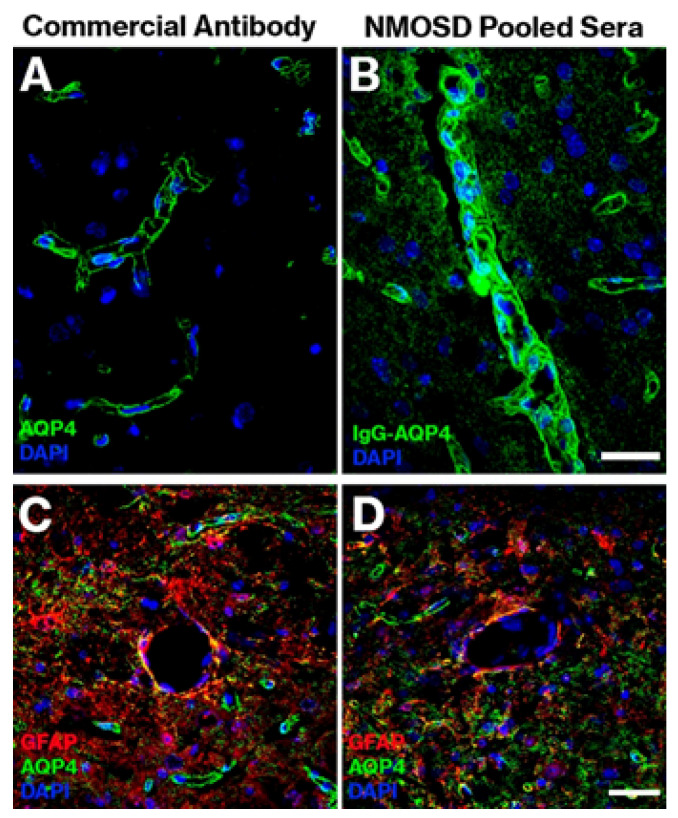
Panels (**A**,**B**): comparison of the labeling observed with commercial anti-AQP4 antibody (dilution 1:100) (**A**) and with the pooled NMOSD sera (IgG–AQP4) (**B**). Panels (**C**,**D**) show confocal microscopy images confirming the presence of AQP4–IgG in mouse brain tissue and its colocalization with GFAP and anti-AQP4, with similar patterns of immunostaining for commercial anti-AQP4 antibody (**C**) and pooled NMOSD patient sera (**D**). Scale bar = 50 μm.

**Figure 3 ijms-22-05192-f003:**
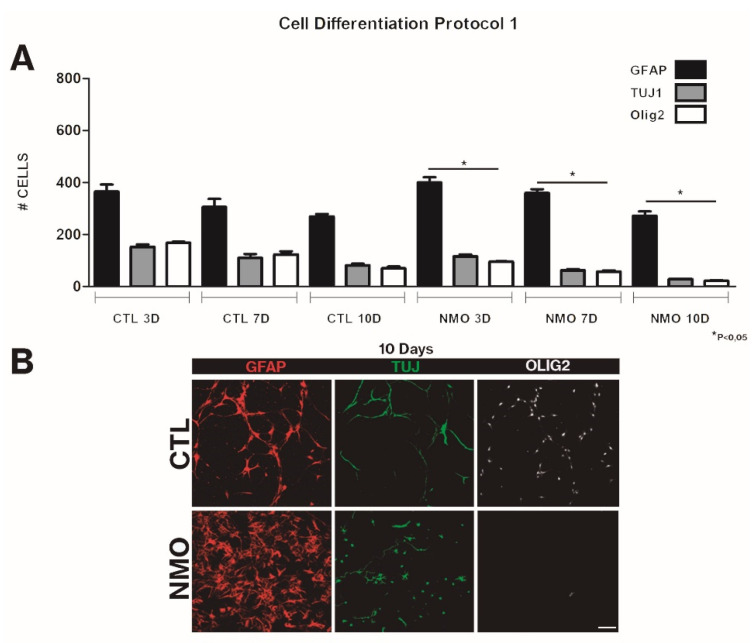
Results obtained for cell differentiation. Graphs of the quantifications at 3, 7, and 10 days after differentiation (**A**). When serum from cases and controls was applied after adding the differentiation medium (protocol 1), we observed a slight increase in gliogenesis (80%)**,** but also decreased differentiation to neurons and oligodendrocytes. In cultures with serum from healthy group, we observed no significant differences between assessments, with proliferation rates similar to baseline values. Images of the neurosphere assay cultures (10 days) (**B**). It is clear that adding serum from patients with NMOSD affects the morphology of neurospheres, which lose their characteristic spherical shape. Astrocytes treated with serum from patients with NMOSD presented a fibrous, protoplasmatic morphology; neural cells showed limited projections. The graph shows means and standard errors. * Significant differences between the NMOSD group and the healthy group (*p* < 0.05). Scale bar = 50 μm.

**Figure 4 ijms-22-05192-f004:**
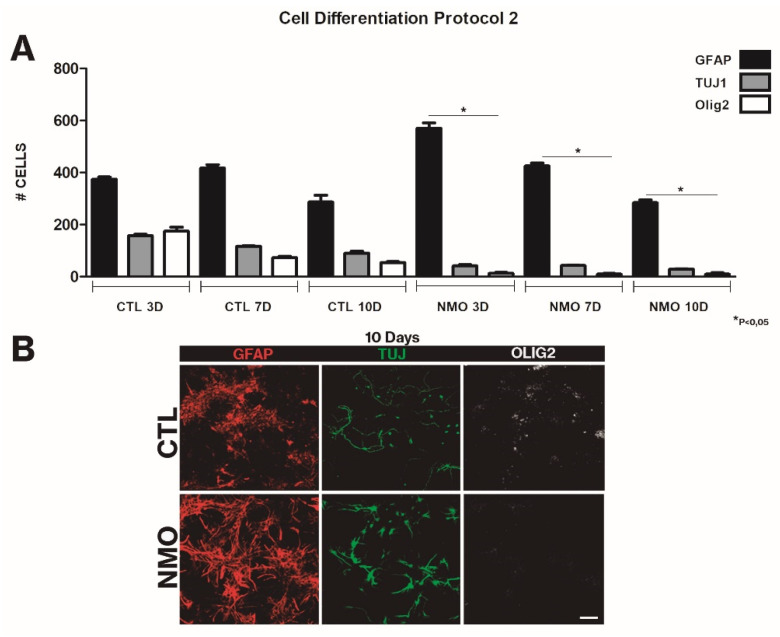
Results obtained for cell differentiation. Graphs of the quantifications at 3, 7, and 10 days after differentiation (**A**). When serum from cases and healthy group was added at the same time as the first differentiation medium (protocol 2), we observed a marked increase in gliogenesis (<80%), but also decreased differentiation to neurons, with values below 10% for TUJ1-positive cells, and reduced differentiation to oligodendrocyte lineage cells, with values below 5%. Images of the neurosphere assay cultures (10 days) (**B**). In cultures treated with serum from controls, we observed no significant differences between assessments, with proliferation rates similar to baseline values. In comparison with protocol 1, the addition of serum from patients with NMOSD at early stages of differentiation clearly induced significant gliogenesis and drastically decreased rates of differentiation to neurons and oligodendrocytes. Furthermore, as in protocol 1, neurospheres presented altered morphology, losing their characteristic spherical shape and showing absence of a defined structure. In the analysis of morphology after differentiation, astrocytes treated with serum from patients with NMOSD showed a fibrous, protoplasmatic morphology; neural cells showed limited projections. The graph shows means and standard error. * Significant differences between the NMOSD group and the healthy group (*p* < 0.05). Scale bar = 50 μm.

**Figure 5 ijms-22-05192-f005:**
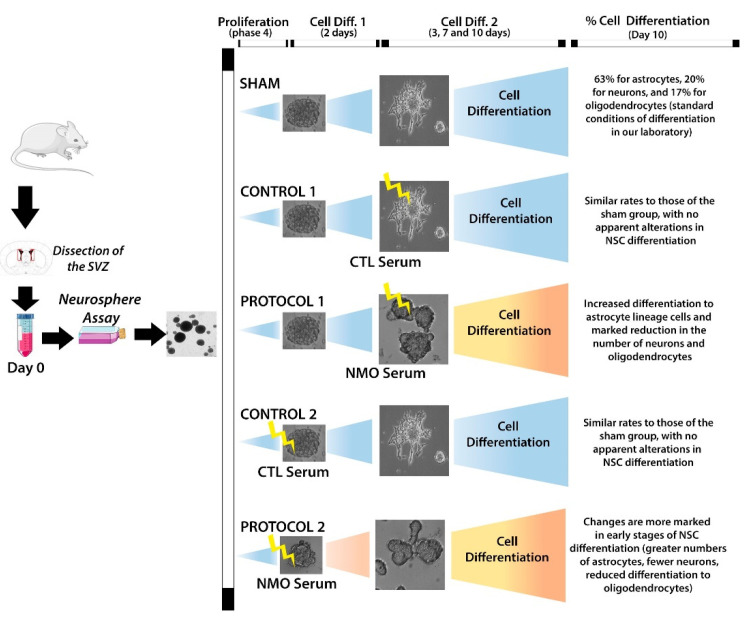
Graphical representation of the hypothesis on the effect of AQP4–IgG on stem cell differentiation, clearly showing that their deleterious effect on differentiation is more pronounced in more immediate precursors.

## Data Availability

The management of patient data will be in compliance with Regulation (EU) 2016/679 of the European Parliament and of the Council, of April 27, 2016, regarding the protection of natural persons and Organic Law 3/2018, Protection of Personal Data and guarantee of digital rights, in force since December 7, 2018. Patient data will be encrypted and will be archived with protection so that only researchers will have access to them. Processing will be done exclusively by authorized individuals. Corrections by these same individuals will be allowed. The datasets used and/or analysed during the current study are available from the corresponding author on reasonable request.

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
