# Peer review of "Sera from Patients with NMOSD Reduce the Differentiation Capacity of Precursor Cells in the Central Nervous System"

_ijms, 2021, doi:10.3390/ijms22105192_

Round 1

Reviewer 1 Report

None of the suggested experiments have been addresses.

The manuscript does not be accepted.

Author Response

Thanks for your comments.

Reviewer 2 Report

Dear Authors,

This is an interesting study.

The major and fundamental issue is that the authors do not consider that NMOSD patients’ sera contain not only AQP4-autoantibody but also some other autoantibodies such as GRP78 autoantibody described reference 3 that is cited in this article. The authors confirmed the presence of AQP4 autoantibody in NMOSD sera by ELISA but it does not rule out the possible presence of other autoantibody. Actually when the authors incubated NMOSD sera with mouse brain sections, labeling was not restricted to the astrocytes but also seen in vascular endothelial cells and pericytes (Figure1 and lines 146-149 in the page 3). Labeling on the vascular endothelial cells was probably not detected by AQP4 antibody but detected by some other antibodies such as GPR78. If the authors would like to show the effect of AQP4-IgG, they must use affinity-purified AQP4 antibody instead of serum.

Lines 34-35 in the page1: “…they have also shown anti-vascular endothelial antibodies, closely with the control of water channels (3)…” What does this sentence mean? Anti-vascular antibodies, which are GRP78 antibodies, cause the increase of BBB permeability; however, this is not related to water channels. Shimizu et al (Sci Transl Med 2017; 9(397): doi:10.1126/scitranslmed.aai9111) suggested that the increased BBB permeability caused by GRP78 results in the invasion of AQP4 autoantibody into the CNS, that facilitates NMO attack.

Lines 215-216 in the page 6: “…it is only observed in the cytoplasm in pathological situations due to the presence of AQP4-IgG (22), as in our study.” This reviewer does not find such a description in the reference 22.

Thank you very much. 

Author Response

Dear Reviewer 2,

Thanks for your comments.

RESPONSE TO COMMENTS

Dear Authors,

This is an interesting study.

The major and fundamental issue is that the authors do not consider that NMOSD patients’ sera contain not only AQP4-autoantibody but also some other autoantibodies such as GRP78 autoantibody described reference 3 that is cited in this article. The authors confirmed the presence of AQP4 autoantibody in NMOSD sera by ELISA but it does not rule out the possible presence of other autoantibody. Actually when the authors incubated NMOSD sera with mouse brain sections, labeling was not restricted to the astrocytes but also seen in vascular endothelial cells and pericytes (Figure1 and lines 146-149 in the page 3). Labeling on the vascular endothelial cells was probably not detected by AQP4 antibody but detected by some other antibodies such as GPR78. If the authors would like to show the effect of AQP4-IgG, they must use affinity-purified AQP4 antibody instead of serum.

Response: Thank you for your comment and observations. We do not discard at any moment the presence of other autoantibodies in the serum of our patients; we carried out other determinations, with findings in two patients (Supplementary Material Table 1). It was not possible for us to carry out the quantification of GRP78 at this particular time, but we are considering detecting said marker for a second phase of the project.

With regard to the comment about sections of rat brain tissue in Figure 1, it is true that markings in other cellular structures are observed beyond foot terminal astrocytes. To confirm the specificity of patients’ serum (Figure 1), we carried out an IHQ with a commercial antibody against AQP4 (Millipore AB3594), adding to this IHQ an antibody specifically for intermediate filaments expressed broadly in astrocytes (GFAP Ck AB4674); in the images obtained, we found marking patterns that were very similar between the serum pool and the purified commercial antibody (Figure 2, C and D), with a high index of colocalization.

COMMENTS

The page1: “…they have also shown anti-vascular endothelial antibodies, closely with the control of water channels (3)…” What does this sentence mean? Anti-vascular antibodies, which are GRP78 antibodies, cause the increase of BBB permeability; however, this is not related to water channels. Shimizu et al (Sci Transl Med 2017; 9(397): doi:10.1126/scitranslmed.aai9111) suggested that the increased BBB permeability caused by GRP78 results in the invasion of AQP4 autoantibody into the CNS, that facilitates NMO attack.

Response: Thank you for your comment. In view of your observation, we have modified the text to “closely with the control of blood brain barrier (BBB) permeability (3)”.

COMMENTS

Lines 215-216 in the page 6: “…it is only observed in the cytoplasm in pathological situations due to the presence of AQP4-IgG (22), as in our study.” This reviewer does not find such a description in the reference 22.

Response: Thank you for your observation. We have eliminated this comment in the text and it now appears as “As it is a transmembrane protein, it is expressed on the surface of cells (…”.

Reviewer 3 Report

Dear Editor, 

The manuscript by Gómez-Pinedo investigates the effect of AQP4-IgG on precursor cells in the CNS. Their results indicate that its effect differs according to the degree of differentiation, and probably affects oligodendrocyte progenitor cells from NG2 cells to a lesser extent than cells from the subventricular zone.

The design of the study and the technical quality of the work look convincing and results can be of general interest. Having human samples is an important factor for the novelty of this work. However, there is a number of major and minor points that would need to be addressed in order to improve the quality of this paper before it can be accepted for publication: 

General: 
 -Defining abbreviation whenever they are firstly introduced and keep using them throughout. For example: line 48 blood-brain barrier and line 51 central nervous system. 
 -Authors are encouraged to use more updated references. I have made some suggestions but this needs to be applied throughout the manuscript; whenever possible. 

Major: 
 The manuscript needs a better structure and a more careful scrutiny by the authors. I will try to break down my comment into smaller points so they can be easier to follow: 

1-Different fonts, colours and referencing styles. All of these need to be addressed. 

2-The introduction is short and doesn’t provide enough foundation knowledge. Authors need to start with a general overview about NMO and its epidemiology (for example) to give a better sense regarding why this is a serious problem. Reference:

https://pubmed.ncbi.nlm.nih.gov/32670177/   https://pubmed.ncbi.nlm.nih.gov/26705758/  

Then, they need to mention that AQPs have been validated as an important drug target but there is no single drug that has yet been approved to successfully target it. This needs to be mentioned, references to be included:

https://www.ncbi.nlm.nih.gov/pmc/articles/PMC4067137/ https://www.ncbi.nlm.nih.gov/pmc/articles/PMC6480248/ 

This will provide a nice build up to identify the gap in knowledge where the authors need to state the reason and the novelty of current study in a little bit more details rather than just “In this article, we present our analysis of the latter point”. 

3-In line 56, it has been mentioned that the control group consist of 3 patients. However in line 59, it has been indicated that they are seven! Which one is correct? Please clarify.

4-Line 82-83: “All participants (cases and controls) agreed to participate in the study and signed informed consent forms”. This statement doesn’t belong here and it should be moved up along with the sample description.

5-Imaging was an essential aspect of this manuscript. Author needs to provide more details such as how many FOVs have been taken and what are their measures to minimize biases, and how they have excluded any possible interference from background signals in order to enhance the reproducibility of the presented data. Magnification number isn’t enough as it has nothing to do with resolution especially for the purpose of quantitative analyses like in this study. So, author needs to include NA of the utilized lens as well. 

6-Authors need to indicate the confluency at which they culture their cells and their doubling time since this can interfere with the interpretation of results if it hasn’t been kept consistent throughout the study. Moreover, authors need to indicate what passage range they have used for their cells in this study. This can be important to determine if the cells kept the phenotypic signature and how they have tested for this.

7-A statement regarding their measures to investigate mycoplasma contamination needs to be mentioned as this can significantly interfere with the conclusion of this study.

8-Re-write the figure legends to only provide the essential information. Antibodies dilution factors etc don’t necessarily belong here.

9-The discussion needs a lot of improvement. First of all, AQP4 is the main focus of this study and hence the discussion should cover some general background about AQP including the following points: 

AQPs are historically known to be passive transporters of water. Lines of evidence in the last decade have highlighted the diverse function of AQPs beyond water homeostasis. Authors need to cover this point. A reference to be included: 

https://www.ncbi.nlm.nih.gov/pubmed/26365508 https://pubmed.ncbi.nlm.nih.gov/29503618/  

Moreover, a subgroup of AQP water channels also facilitates transmembrane diffusion of small, polar solutes not only water; aquaglyceroporin. References to be included: 

https://www.ncbi.nlm.nih.gov/pubmed/16715408 https://www.ncbi.nlm.nih.gov/pubmed/31889130  https://www.ncbi.nlm.nih.gov/pmc/articles/PMC6048697/  

Line 219-220: “As it is a transmembrane protein, it is expressed on the surface of cells”. The increased AQP4 expression and the redistribution/surface localization can be two different concepts. Previous studies have shown an increased in AQP4 membrane localisation in primary human astrocytes which wasn’t accompanied by a change in AQP4 protein expression levels. This mislocalization can be a potential therapeutic target. References: •  

https://www.ncbi.nlm.nih.gov/pmc/articles/PMC5765450/ https://www.ncbi.nlm.nih.gov/pubmed/31242419 https://pubmed.ncbi.nlm.nih.gov/23505074/

Recent studies have validated that targeting AQP4 is a viable therapeutic target for various CNS injuries including TBI and stroke. Kitchen et al Cell 2020 has nicely shown that inhibition AQP4 localization to the blood-spinal cord barrier, ablated CNS edema, and led to accelerated functional recovery compared with untreated animals. These results have been proven to valid for stroke as well as indicated by Sylvain et al BBA 2021. Targeting this subcellular localisation of AQP4 might be a future therapeutic option where the author can investigate in their novel model. References to be included:

https://pubmed.ncbi.nlm.nih.gov/32413299/

https://pubmed.ncbi.nlm.nih.gov/33561476/

Secondly, it was very nice to see that authors have indicated some of the limitations of the current study. Authors also need to discuss future directions which can benefit from 3D self-organized models and human brain organ-on-a-chip platforms, especially those amenable for advanced imaging to monitor the mediators involved in the process of neuroinflammation in real-time and also TEM. References to be included:

https://pubmed.ncbi.nlm.nih.gov/25033469/  -https://pubmed.ncbi.nlm.nih.gov/33117784/  
https://pubmed.ncbi.nlm.nih.gov/30165870/ 

Lastly and in lines 279-280 “At this point, a potential synergy between cell therapy and bioengineering may involve the use of biomaterials with biomedical applications in the central nervous system”. Authors need to introduce other methodologies for target identification and validation; particularly, the increased use of high-throughput screening. This has been nicely reviewed recently by Aldewachi et al 2021 and also Del Palacio et al 2016. References to be included:

https://pubmed.ncbi.nlm.nih.gov/33672148/ https://pubmed.ncbi.nlm.nih.gov/26962874/ https://www.ncbi.nlm.nih.gov/pmc/articles/PMC3336779/  

Best. 

Author Response

Dear Reviewer 3,

Thanks for your comments

COMMENTS

The manuscript by Gómez-Pinedo investigates the effect of AQP4-IgG on precursor cells in the CNS. Their results indicate that its effect differs according to the degree of differentiation, and probably affects oligodendrocyte progenitor cells from NG2 cells to a lesser extent than cells from the subventricular zone.

The design of the study and the technical quality of the work look convincing and results can be of general interest. Having human samples is an important factor for the novelty of this work. However, there is a number of major and minor points that would need to be addressed in order to improve the quality of this paper before it can be accepted for publication:

Response: We appreciate the critical and constructive character of your comments, which will be annexed to the document.

COMMENTS

General:

 -Defining abbreviation whenever they are firstly introduced and keep using them throughout. For example: line 48 blood-brain barrier and line 51 central nervous system. 

Response: Thank you for the observation. We proceeded to revise the document and correct said point.

 -Authors are encouraged to use more updated references. I have made some suggestions but this needs to be applied throughout the manuscript; whenever possible. 

Response: Thank you for your input regarding our references. We proceeded to apply your suggestions.

COMMENTS

Major: 

The manuscript needs a better structure and a more careful scrutiny by the authors. I will try to break down my comment into smaller points so they can be easier to follow: 

Response: Thank you for your comments. We tried to improve the structure of the document by following your recommendations.

1-Different fonts, colours and referencing styles. All of these need to be addressed. 

Response: Thank you. We revised the document in full in order to correct fonts, colors, and referencing styles.

2-The introduction is short and doesn’t provide enough foundation knowledge. Authors need to start with a general overview about NMO and its epidemiology (for example) to give a better sense regarding why this is a serious problem. Reference:

https://pubmed.ncbi.nlm.nih.gov/32670177/

https://pubmed.ncbi.nlm.nih.gov/26705758/

Response: Thank you. We have revised the introduction, which appears as follows. 

Then, they need to mention that AQPs have been validated as an important drug target but there is no single drug that has yet been approved to successfully target it. This needs to be mentioned, references to be included:

https://www.ncbi.nlm.nih.gov/pmc/articles/PMC4067137/

https://www.ncbi.nlm.nih.gov/pmc/articles/PMC6480248/

This will provide a nice build up to identify the gap in knowledge where the authors need to state the reason and the novelty of current study in a little bit more details rather than just “In this article, we present our analysis of the latter point”. 

Response: Neuromyelitis optica spectrum disorders (NMOSD) is a rare demyelinating neuroinflammatory disease of the central nervous system (CNS) that mainly attacks the optic nerves and the spinal cord, which can lead to vision loss and paralysis (1, 2). A considerable percentage of patients with NMOSD are positive for IgG1 autoantibodies (IgG) targeting the aquaporin-4 (AQP4) water channel protein expressed in astrocytes (AST), and other antibodies may also occur, such as myelin oligodendrocyte glycoprotein (MOG) -antibody- seropositive inflammatory demyelinating disease (3, 4); they have also shown anti-vascular endothelial antibodies closely related with the implication of blood-brain barrier (BBB) ​​permeability (5), resulting in an inflammatory response with complement participation, leading to demyelination (6), and therefore, to neurological sequelae (7). AQP4-antibody disease has an age of onset of 40 years and shows a high female-to-male ratio (up to 9: 1). Today, there is no specific treatment in NMOSD (8,9). The therapeutic approach is based on the use of immunosuppressants, plasmapheresis, and anti-CD19 and anti-CD20 drugs, aiming to reduce the presence of B cells (10,11). Recently, new strategies with promising results in preclinical stages are being evaluated, based on AQP inhibitor chemicals and anti AQP4 monoclonal antibodies (9-10). An important point is that remyelination is thought to be limited in NMOSD, and there are no treatments able to restore myelin; however, the use of drugs that may favor remyelination is being considered in multiple sclerosis (11), and it has also been suggested that they may promote remyelination in NMOSD, as early loss of oligodendrocytes (OLG) has been reported (12,13,14,15). Others authors have reported NMO-IgG could decrease the expression of connexins and cause oligodendrocytic damage and demyelination in NMOSD (16). For instance, clobetasol, which promotes OLG differentiation (17), has been analyzed in an in vivo NMOSD model, with promising results. However, this hypothesis raises unresolved questions about NMOSD: the effect of the altered interaction between ASTs and OLG-AST (18, 19), whether blood-brain barrier rupture in NMOSD may inhibit OLG migration, whether the inflammatory environment in active NMOSD lesions and the altered microglial response may inhibit remyelination and favor irreversible axonal lesion (20), and whether AQP4-IgG may have an effect on cell differentiation in the central nervous system. In this article, we present our analysis of the latter point.

3-In line 56, it has been mentioned that the control group consist of 3 patients. However in line 59, it has been indicated that they are seven! Which one is correct? Please clarify.

Response: Thank you for your observation. We apologize, as this has been a mistake and in fact, we have three healthy control patients instead of seven. We have corrected the text.

4-Line 82-83: “All participants (cases and controls) agreed to participate in the study and signed informed consent forms”. This statement doesn’t belong here and it should be moved up along with the sample description.

Response: Thank you for your suggestion. We have changed this paragraph so that it appears after the authorization of the ethics committee of the Hospital Clínico San Carlos.

5-Imaging was an essential aspect of this manuscript. Author needs to provide more details such as how many FOVs have been taken and what are their measures to minimize biases, and how they have excluded any possible interference from background signals in order to enhance the reproducibility of the presented data. Magnification number isn’t enough as it has nothing to do with resolution especially for the purpose of quantitative analyses like in this study. So, author needs to include NA of the utilized lens as well. 

Response: Thank you. The characteristics of the objectives used are as follows.

UPLXAPO40X , NA 0,95: Used in the quantitative analysis of the cellular cultures.

UPLXAPO20X, NA 0,8: Used for the acquisition of panoramic images for the analysis of rat dissections. The acquisition of images was done by confocal microscopy, using the sequential acquisition mode to avoid interference of tails or heads of the fluorescent spectra of the secondary antibodies used; in addition, the omission of the primary antibody as a negative control was carried out in a chamber according to each time and condition. For the tissues, the same control procedure was used. This point will be annexed in the methods section.

6-Authors need to indicate the confluency at which they culture their cells and their doubling time since this can interfere with the interpretation of results if it hasn’t been kept consistent throughout the study. Moreover, authors need to indicate what passage range they have used for their cells in this study. This can be important to determine if the cells kept the phenotypic signature and how they have tested for this.

Response: Thank you for your comment, it will be attached to the supplementary material

Passage number 5 was chosen for carrying out the experiments, since we know that the phenotypic signature of the neurospheres remains without changes until passage number 10 (ICC and WB, data do not show). The cellular confluence observed was 70% confluence of the neurospheres. The ratio of expansion was a ratio of 1:3-1:5, which is consistent with the references bibliography, considering this as the behavior expected of healthy cells.

7-A statement regarding their measures to investigate mycoplasma contamination needs to be mentioned as this can significantly interfere with the conclusion of this study.

Response: Thank you for your comment. This point has also been annexed in Supplementary Material 3. Before beginning the experiment, we proceeded with mycoplasma analysis, for which a MycoAlert PLUS detection kit was used according to the manufacturer’s protocol in the fifth passage.

8-Re-write the figure legends to only provide the essential information. Antibodies dilution factors etc don’t necessarily belong here.

Response: We modified the figure legend and moved information to the corresponding methods section.

9-The discussion needs a lot of improvement. First of all, AQP4 is the main focus of this study and hence the discussion should cover some general background about AQP including the following points:        

AQPs are historically known to be passive transporters of water. Lines of evidence in the last decade have highlighted the diverse function of AQPs beyond water homeostasis. Authors need to cover this point. A reference to be included:

Response: Thank you for your comment. This point has also been annexed:

In recent years, a new role of AQP4 has been described, specifically its role in the inflammatory response mediating the migration of immune cells, edema and as a marker for a bad prognosis in gliomas (24).

https://www.ncbi.nlm.nih.gov/pubmed/26365508 https://pubmed.ncbi.nlm.nih.gov/29503618/

Response: Thank you for your comment. This point has also been annexed:

Moreover, a subgroup of AQP has been described that also facilitates transmembrane diffusion of small polar solutes, not just water; these are aquaglyceroporins, with a narrow function in lipid metabolism (25, 26), slightly associated with pathologies such as diabetes mellitus type 2.

 References are included.                                                                                            

https://www.ncbi.nlm.nih.gov/pubmed/16715408 https://www.ncbi.nlm.nih.gov/pubmed/31889130

https://www.ncbi.nlm.nih.gov/pmc/ artículos / PMC6048697 /

Line 219-220: “As it is a transmembrane protein, it is expressed on the surface of cells”. The increased AQP4 expression and the redistribution/surface localization can be two different concepts. Previous studies have shown an increased in AQP4 membrane localisation in primary human astrocytes which wasn’t accompanied by a change in AQP4 protein expression levels. This mislocalization can be a potential therapeutic target. References:

Response: Thank you for your comment. We have modified the text as follows.

As it is a transmembrane protein, it is expressed on the surface of cells (28, 29). The presence of a change in the expression of the astrocytes can be considered as a therapeutic target.  Salaman et al., in an in vitro model and using human cortical astrocytes, described that hypothermia can induce an increment of the expression of AQP4; this can be a favorable strategy for the treatment of edema (30).

https://www.ncbi.nlm.nih.gov/pmc/articles/PMC5765450/

https://www.ncbi.nlm.nih.gov/pubmed/31242419 https://pubmed.ncbi.nlm.nih.gov / 23505074 /

Recent studies have validated that targeting AQP4 is a viable therapeutic target for various CNS injuries including TBI and stroke. Kitchen et al Cell 2020 has nicely shown that inhibition AQP4 localization to the blood-spinal cord barrier, ablated CNS edema, and led to accelerated functional recovery compared with untreated animals. These results have been proven to valid for stroke as well as indicated by Sylvain et al BBA 2021. Targeting this subcellular localization of AQP4 might be a future therapeutic option where the author can investigate in their novel model. References to be included:

Response: Thank you for your comment. We have annexed the following paragraph.

Recent experimental studies have correlated that the localized inhibition of AQP4 can favor hastened functional recovery in lesions such as stroke, or traumatic lesions where inflammatory substrate plays an important role (31,32).

Kitchen et al Cell 2020 --Sylvain et al BBA 2021

https://pubmed.ncbi.nlm.nih.gov/32413299/                      

https://pubmed.ncbi.nlm.nih.gov/33561476/

Secondly, it was very nice to see that authors have indicated some of the limitations of the current study. Authors also need to discuss future directions which can benefit from 3D self-organized models and human brain organ-on-a-chip platforms, especially those amenable for advanced imaging to monitor the mediators involved in the process of neuroinflammation in real-time and also TEM. References to be included:

Response: Thank you for your comments and suggestions. We have annexed the following text.

In recent years, the development of devices of 3D culture based on the generation of organoids derived from iPSCs can be of great utility in advancing the generation of biohybrid devices to evaluate new therapeutic alternatives, further the comprehension of molecular transport mechanisms, and providing an individualized model to simulate conditions close to those that could be found in actual patient tissue (52, 53).

https://pubmed.ncbi.nlm.nih.gov/25033469/

https: //pubmed.ncbi.nlm.nih.gov/33117784/

https://pubmed.ncbi.nlm.nih.gov/30165870/

Lastly and in lines 279-280 “At this point, a potential synergy between cell therapy and bioengineering may involve the use of biomaterials with biomedical applications in the central nervous system”. Authors need to introduce other methodologies for target identification and validation; particularly, the increased use of high-throughput screening. This has been nicely reviewed recently by Aldewachi et al 2021 and also Del Palacio et al 2016. References to be included:

Response: Thank you for your comment. We consider this to be a fundamental point in the ability to generate devices for the application of high throughput screening techniques, generating libraries specific for screening by way of biochemical assays and cell-bases assay.

The following paragraph is annexed.

The development of techniques of high throughput screening (HTS) techniques can be a complementary tool, as well as assays based on cells and biochemical techniques such as those described by Aldewachi et al. in 2021 and by Del Palacio et al. in 2016, which are necessary and fundamental for the analysis of the potential of possible therapeutic drugs (54, 55). 

https://pubmed.ncbi.nlm.nih.gov/33672148/ https://pubmed.ncbi.nlm.nih.gov/26962874/ https://www.ncbi.nlm.nih.gov/pmc/articles/ PMC3336779 /

Round 2

Reviewer 1 Report

The paper achive the basic level to be published.
Some issues that i have suggested does not be addressed. However the paper contains interesting findings.

Minor ponits.

The references must be improved.

Please add these references:

Page 1, lane 36: add between the references original articles in which AQP4 targeting was charachterized (PMID: 21212277; PMID: 19229993).

Page 8, lane 241: add between the references the original article (PMID: 30877104).

Author Response

The paper achive the basic level to be published.
Some issues that i have suggested does not be addressed. However the paper contains interesting findings.

Response: We appreciate the comments made and your critical view

Minor ponits.

The references must be improved.

Please add these references:

Page 1, lane 36: add between the references original articles in which AQP4 targeting was charachterized (PMID: 21212277; PMID: 19229993).

Page 8, lane 241: add between the references the original article (PMID: 30877104).

Response: Thank you for your suggestions and we attach the requested references, we consider that they are transcendent in our manuscript.

Attached version of the manuscript with the change control

Reviewer 2 Report

Dear Authors,

This reviewer would point out the same issue as the round 1.

The authors examined the effects of patients’ sera, which probably include AQP4 autoantibodies. If the authors want to clarify the effect of AQP4 autoantibodies themselves, it is required to use the affinity-purified AQP4 autoantibodies instead of sera. Otherwise, the results cannot exclude the possible effects of some other antibodies or molecules in the sera.

The title of this study should be, for example, “NMOSD patients sera reduce the differentiation capacity of precursor cells in the central nervous system” In addition, most of “AQP4-IgG” in the text should be appropriately replaced with something like “patients’ sera”.

sham group (in the line 73 and 166): This reviewer does not think these are “sham group” in this case, because these seem to be an immunohistochemical control in which the primary antibody incubation was omitted.

Line 182 in the legend of Figure 1: The authors described that patients’ sera showed positive labeling in pericytes surrounding capillaries. It is required to confirm this by using some markers for pericytes. This reviewer thinks that the labeling in the Figure 1C is probably on the endfeet of astrocytes surrounding capillaries.

Supplementary Material 2: The authors described that serum was diluted in 0.1 M paraformaldehyde (Sigma P6148). Did the authors really dilute the antibody in 0.1 M paraformaldehyde?

Thank you very much.

Author Response

This reviewer would point out the same issue as the round 1.

The authors examined the effects of patients’ sera, which probably include AQP4 autoantibodies. If the authors want to clarify the effect of AQP4 autoantibodies themselves, it is required to use the affinity-purified AQP4 autoantibodies instead of sera. Otherwise, the results cannot exclude the possible effects of some other antibodies or molecules in the sera.

Response: thank you for your valuable comments, one of our limitations is that it was not possible to purify and we are aware of the possible interactions of other antibodies or undetermined or unknown molecules. The following text is attached in the study limitations section: 

                        Our study presents some limitations. Although the data presented show a considerable difference between the NMOSD group and the control group, with serum from patients with NMOSD having a clear effect on neural differentiation, we must acknowledge that the sample size is small and our serum samples may contain other antibodies or undetermined or unknown molecules; future studies should use larger sample sizes and quantify the antibody in order to establish a correlation with clinical and imaging profiles

The title of this study should be, for example, “NMOSD patients sera reduce the differentiation capacity of precursor cells in the central nervous system” In addition, most of “AQP4-IgG” in the text should be appropriately replaced with something like “patient’s sera”.

Response: thank you for your suggestion, it is valuable and we will attach it to the document

sham group (in the line 73 and 166): This reviewer does not think these are “sham group” in this case, because these seem to be an immunohistochemical control in which the primary antibody incubation was omitted.

Response: Thank you for your comment, I think it is a confusion in the wording, we will review it to make it clear that we have a sham group (for cell culture protocols) and also controls in the IHC procedures  

In the line 67 (before 66), the following text is attached:

…..and a sham group, for which the sample included no serum (only culture medium without human sera)……..

In the line 158 (Before 166), the following text is attached:

AQP4-IgG detection. Increased detection of AQP4-IgG was observed in tissue treated with samples from the NMOSD group (high levels of fluorescent signal), although intensity varied between cases; very slight levels were observed in controls, and no labeling was observed in the sham group (Figure 1-A)(tissue incubated without any serum).Labeling was observed in all the sections analyzed, regardless of the anatomical region (cortex, cerebellum, hypothalamus, etc). By location, labeling mainly corresponded to glial cells (astrocytes and astrocytic end foot) (Figure 1-B-C).

Line 182 in the legend of Figure 1: The authors described that patients’ sera showed positive labeling in pericytes surrounding capillaries. It is required to confirm this by using some markers for pericytes. This reviewer thinks that the labeling in the Figure 1C is probably on the endfeet of astrocytes surrounding capillaries.

Response: Thank you for your comment and if we have the corresponding image acquired by confocal microscopy, only that we had not presented it, it is attached for your consideration (See Image 1 and 2 only for reviewer)

Supplementary Material 2: The authors described that serum was diluted in 0.1 M paraformaldehyde (Sigma P6148). Did the authors really dilute the antibody in 0.1 M paraformaldehyde?

Response: sorry is an error that we have already corrected, an apology

Attached version of the manuscript with the change control

Thank you very much

Reviewer 3 Report

Dear Editor,

The authors have successfully addressed the majority of my comments and concerns in order to improve the quality of the manuscript.

I do believe that the corrections, additional sections and updated references, have contributed to enhancing the clarity of the manuscript, which I can now endorse for publication.

All the best!

Author Response

Dear Editor,

The authors have successfully addressed the majority of my comments and concerns in order to improve the quality of the manuscript.

I do believe that the corrections, additional sections and updated references, have contributed to enhancing the clarity of the manuscript, which I can now endorse for publication.

All the best!

Response: Dear Reviewer We appreciate your constructive criticism

Round 3

Reviewer 2 Report

I do not understand what the Images 1 and 2 mean.  I guess the authors want to show that the labeling are seen in pericytes. I do not completely agree this, but I have no more comments. 

Thank you very much.  

Author Response

Q1: I do not understand what the Images 1 and 2 mean. I guess the authors want to show that the labeling are seen in pericytes. I do not completely agree this, but I have no more comments.

Thank you very much

Response: Thanks for your comments

Serum from NMOSD patients colocalize IHC with pericyte markers

In the analysis of the immunostaining using the group of sera from patients with NMOSD, expression is observed in astrocytes and cells that apparently could correspond to pericytes. In order to verify this point, a double immunostaining was performed for the specific marking of pericytes, using the NG2 antibody already described in the literature as a pericyte marker (1-3). Commercial antibody against NG2 (MS, NG2 Millipore MAB5384, 1: 200) was used, with which we performed an IHC and subsequently analyzed the samples with a confocal microscope. Olympus AF1000 microscope software, using a sequential acquisition protocol with a resolution of 800 px, at 4 msec x pixel, with a magnification of 60X (NA 0.84); regions of interest were identified for further analysis of co localitation. Image A and B show the colocalization of the antibody with the serum of patients with NMOSD.

Image A

Confocal microscopy image acquired at 60x with a 2x digital zoom, in channel-by-channel mode (sequential mode, 800 x 800 pixels). It shows the fluorescence image and the colocalization plot, in the individual panels we observe in A) channel 488 in green NMOSD Sera, B) in channel 555 in red the ac NG2, C) shows the fusion of both channels (red and green) and in D) the image of the colocalization analysis (white doots).

Image B

Confocal microscopy image acquired at 60x with a 2x digital zoom, in channel-by-channel mode (sequential mode, 800 x 800 pixels), with transmitted light image as spatial orienting. It shows the fluorescence image and the colocalization graph, in the individual panels we observe in A) channel 488 in green NMOSD Sera, B) in channel 555 in red the ac NG2, C) shows the fusion of both channels (red and green) and in D) the image of the colocalization analysis (white doots).

References

  1. Yamazaki Tomoko, Mukouyama Yoh-suke. Tissue Specific Origin, Development, and Pathological Perspectives of Pericytes. Frontiers in Cardiovascular Medicine (2018) 5: 78. doi: 10.3389 / fcvm.2018.00078
  2. Ozerdem U, Grako KA, Dahlin-Huppe K, Monosov E, Stallcup WB. NG2 proteoglycan is expressed exclusively by mural cells during vascular morphogenesis. Dev Dyn. (2001) 222: 218-27. doi: 10.1002 / dvdy.1200
  3. Stallcup WB. The NG2 proteoglycan: past insights and future prospects. J Neurocytol. (2002) 31: 423–35. doi: 10.1023 / A: 1025731428581

We attach the file (pdf) of the supplementary material

This manuscript is a resubmission of an earlier submission. The following is a list of the peer review reports and author responses from that submission.

Round 1

Reviewer 1 Report

  1. Resuts:

Paragraph 3.1:

Patients and controls

  1. Patients and controls: authors indicate the presence of AQP4-IgG as “AQP4-IgG expression”. The sentence AQP4-expression, must be changed in AQP4-IgG detection.

  1. The authors use immunofluorescence on mouse brain sections to confirm the presence of AQP4-IgG in patient sera. The authors report the sentence: “By location, labeling mainly corresponded to cells from the vascular endothelium, pericytes surrounding capillaries, although it was also observed in cells adjacent to the ventricular wall, as well as in some unidentified cells (possibly astrocytes)”. This sentence is completely wrong. Authors must change the sentence accordingly to data reported in the paper (PMID: 24260168) that must be cited into the text. Notably, AQP4 is exclusively expressed by astrocytes, therefore indicate endothelial or pericytes as potential cells in which AQP4 is expressed is completely wrong.

Paragraph 3.2:

Effect on SVZ cell differentiation

  1. The study was conducted using 2% human serum in the primary culture of neurospheres from the subventricular zone. Authors observe an alteration of cell differentiation. Authors conclude that AQP4-IgG affects cell differentiation. Despite the basic idea is good, to conclude that the effect is due to AQP4-IgG, authors must repeat the same experiment using purified AQP4-IgG from NMO serum. This is a crucial experiment, because NMO serum contains other molecules potentially able to induces the same effect on SVZ differentiation. To purify AQP4-IgG authors must follow method reported in the paper (PMID: 25239624). This must be cited into the text.

No data is reported about the effect of NMO-serum or purified AQP4-IgG on astrocyte cell vitality. I think that the reduction of astrocyte differentiation could be associated to the AQP4-IgG attack to AQP4 expressed by astrocyte. This induce a well-known complement dependent and independent cytotoxicity (CDC). Authors must check whether the CDC is involved as reported (PMID: 25239624).

Discussion

The discussion must be tuned-down because is excessive speculative. And must be aligned to data obtained using purified AQP4-IgG.

Reviewer 2 Report

Dear Authors,

This is an interesting study.

The major and fundamental issue is that the authors do not consider that NMOSD patients’ sera contain not only AQP4-autoantibody but also some other autoantibodies such as GRP78 autoantibody described reference 3 that is cited in this article. The authors confirmed the presence of AQP4 autoantibody in NMOSD sera by ELISA but it does not rule out the possible presence of other autoantibody. Actually when the authors incubated NMOSD sera with mouse brain sections, labeling was not restricted to the astrocytes but also seen in vascular endothelial cells and pericytes (Figure1 and lines 146-149 in the page 3). Labeling on the vascular endothelial cells was probably not detected by AQP4 antibody but detected by some other antibodies such as GPR78. If the authors would like to show the effect of AQP4-IgG, they must use affinity-purified AQP4 antibody instead of serum.

Lines 34-35 in the page1: “…they have also shown anti-vascular endothelial antibodies, closely with the control of water channels (3)…” What does this sentence mean? Anti-vascular antibodies, which are GRP78 antibodies, cause the increase of BBB permeability; however, this is not related to water channels. Shimizu et al (Sci Transl Med 2017; 9(397): doi:10.1126/scitranslmed.aai9111) suggested that the increased BBB permeability caused by GRP78 results in the invasion of AQP4 autoantibody into the CNS, that facilitates NMO attack.

Lines 215-216 in the page 6: “…it is only observed in the cytoplasm in pathological situations due to the presence of AQP4-IgG (22), as in our study.” This reviewer does not find such a description in the reference 22.

Thank you very much. 

Reviewer 3 Report

In the present paper, Gómez-Pinedo and co-workers have assessed the differentiation capacity of neurospheres from the subventricular zone of mice by adding serum from Neuromyelitis optica spectrum disorders. The experiments were performed at two different time points: at an early and at an advanced stage of differentiation.

Unfortunately, at this stage the study appears to be the work in progress. I would kindly suggest to the authors that they first perform some additional experiments; perhaps on isolated single cell types that they assessed, which would focus on mechanistic action of NMO antibodies. The manuscript also fails to clearly show which groups in the results were tested (e.g. in Figure 3 the authors show significant differences only between GFAP and Olig2 groups in NMO treatments. This is unusual, since TUJ1 group is usually almost at the same level as Olig2. Furthermore, graphs do not show that NMO treatment vs control (I assume that CTL denotes control?) was tested.)

In addition, in many parts the manuscript is very difficult to follow and needs to be thoroughly rewritten. I strongly encourage the authors to send the next draft of the manuscript to a language–editing service.

Reviewer 4 Report

The manuscript presented by Gómez-Pinedo et al, explore a very interesting issue, minimally explored by now yet, consistent in investigate the role that antibodies against AQP4, as those present in the serum of patients with NMOSD, may produce over the differentiation capacity in culture of neurospheres obtained from the SVZ of mice.

The work is properly written, experiments are adequate to address the questions they want to answer and in general the conclusions rise by authors can be sustained by the results obtained. However, I have several concerns regarding how the results are presented and the lack of statistical analysis for some experimental groups compared. Moreover, I believe some changes in the discussion section and the conclusions might indicate more precisely the results obtained by authors in this work.

Concerns:

  1. ELISA kits to evaluate positiveness of IgG-anti AQP4 in serum of patients is not the most reliable system for such goal, as abundantly indicated by many authors in the literature. Cell based assay ((CBA) system is much more efficient and specific, and does not produce false positive. It is not clearly indicated if the serum of patients was individually assayed by IHC systems, and whether or not a pool of serum was used for the neurosphere assays.

  1. It is really incredible that all experiments were just performed with only 3 animals, and using the antiserum from only six patients NMOSD(+), from who levels of IgG-anti AQP4 were not measured. Could authors comment on this highly efficiency procedure?

  1. In the results shown in Figure 3 and Figure 4, the statistical analysis presented corresponds for the comparison of the proliferation (# cells) of the three different type of cells; astrocytes, neurons and oligodendrocytes, under one condition, either control (CTL) or treated with anti-NMOSD serum at different times of culture development. But, in order to evaluate the effect that NMOSD serum is producing in the neurosphere development, the interesting analysis would be to compare rate of proliferation between CTL and NMOSD-treated, per each cell type at each time of development, after treatment. In other words, the growth (# cells) now just show differences among cell types comparing NMOSD treated cells among them, or CTL treated cells among them, but no statistical comparison is shown per cell type between CTL and NMOSD. I think these comparisons are necessary, to rise the conclusions that the authors indicate in this work.

  1. In the Discussion, at the beginning of this section authors say that AQP4 is only observed in the cytoplasm in pathological situations due to the presence of AQP4-IgG, as in their study. That sentence is not fully true. For instances, it is been recently indicated that shuttling of AQP4 from intracellular vesicles to the cell membrane under specific stimuli is an absolutely normal physiological mechanism (See, Kitchen P. et al, Cell 2020. 181 (4): 784-799).

  1. In the Discussion, the authors numerated several conclusions. Under the third conclusion, authors indicate that: “cell therapy with OPCs, which may protect against AQP4-IgG, seems to be the most probable target in the research of a restorative therapy for NMOSD. This phrase, at this time of the work presented, is more a working hypothesis yet. No experiments have been done in the presented work to rise this conclusion.

  1. I would strongly recommend that to rise the most important conclusion authors want to propose, the statistical analysis comparing groups of CTL vs. NMOSD cells must be done and shown. Therefore, Conclusions could probably be rephrased accordingly such analysis.